# Cell-Free DNA-Methylation-Based Methods and Applications in Oncology

**DOI:** 10.3390/biom10121677

**Published:** 2020-12-15

**Authors:** Francesca Galardi, Francesca De Luca, Dario Romagnoli, Chiara Biagioni, Erica Moretti, Laura Biganzoli, Angelo Di Leo, Ilenia Migliaccio, Luca Malorni, Matteo Benelli

**Affiliations:** 1«Sandro Pitigliani» Translational Research Unit, Hospital of Prato, 59100 Prato, Italy; francesca.galardi@uslcentro.toscana.it (F.G.); francesca.deluca@uslcentro.toscana.it (F.D.L.); ilenia.migliaccio@uslcentro.toscana.it (I.M.); luca.malorni@uslcentro.toscana.it (L.M.); 2Bioinformatics Unit, Hospital of Prato, 59100 Prato, Italy; dario.romagnoli@uslcentro.toscana.it (D.R.); chiara.biagioni@uslcentro.toscana.it (C.B.); 3«Sandro Pitigliani» Medical Oncology Department, Hospital of Prato, 59100 Prato, Italy; erica.moretti@uslcentro.toscana.it (E.M.); laura.biganzoli@uslcentro.toscana.it (L.B.); angelo.dileo@uslcentro.toscana.it (A.D.L.)

**Keywords:** DNA-methylation, epigenetics, liquid biopsy, cell-free DNA, oncology, precision medicine, genomics, bioinformatics, sequencing

## Abstract

Liquid biopsy based on cell-free DNA (cfDNA) enables non-invasive dynamic assessment of disease status in patients with cancer, both in the early and advanced settings. The analysis of DNA-methylation (DNAm) from cfDNA samples holds great promise due to the intrinsic characteristics of DNAm being more prevalent, pervasive, and cell- and tumor-type specific than genomics, for which established cfDNA assays already exist. Herein, we report on recent advances on experimental strategies for the analysis of DNAm in cfDNA samples. We describe the main steps of DNAm-based analysis workflows, including pre-analytics of cfDNA samples, DNA treatment, assays for DNAm evaluation, and methods for data analysis. We report on protocols, biomolecular techniques, and computational strategies enabling DNAm evaluation in the context of cfDNA analysis, along with practical considerations on input sample requirements and costs. We provide an overview on existing studies exploiting cell-free DNAm biomarkers for the detection and monitoring of cancer in early and advanced settings, for the evaluation of drug resistance, and for the identification of the cell-of-origin of tumors. Finally, we report on DNAm-based tests approved for clinical use and summarize their performance in the context of liquid biopsy.

## 1. Introduction

Liquid biopsy is emerging as a powerful tool for the non-invasive detection of cancer, providing clinically valuable information on patient outcome and informing treatment decisions. The most successful use of liquid biopsy is the analysis of tumor DNA from cell-free DNA (cfDNA) fragments released into the bloodstream through apoptosis, necrosis, and/or active secretion. These fragments can be contained in blood, urine, saliva, cerebrospinal fluid, and stool. The main applications of liquid biopsy include the genomic characterization of cancer, the detection and quantification of the disease in patients with cancer, and the evaluation of treatment response [1,2,3].

Recently, many works have focused on the analysis of DNA-methylation (DNAm) in cfDNA samples. DNAm is a key player in gene regulation, and is one of the most studied epigenetic mechanisms. Its role in cancer initiation, as well as progression and resistance to targeted therapies has been extensively investigated [4,5,6]. For these reasons, DNAm-based biomarkers offer great promise for detecting, monitoring, and characterizing disease in different tumor types [7,8,9,10].

Herein, we provide an overview of methods for the evaluation of DNAm from cfDNA samples. We illustrate the main methodological steps and available techniques to process and deal with the peculiarity of these samples, from sample collection to computational data analysis. Then, we report on existing works exploiting DNAm-based cfDNA analyses to detect and monitor cancer in the early and advanced setting, to evaluate drug resistance, and identify the cell-of-origin of tumor. Finally, a description of clinically approved DNAm-based tests is reported.

## 2. Overview on DNA-Methylation Analysis

As schematically represented in Figure 1, the analysis of DNAm from cfDNA can be divided into four main steps including pre-analytics (sample collection and DNA extraction), DNA treatment, DNAm evaluation through tailored assays, and data analysis. The illustrated sections provide a detailed description of these steps. For clarity, in this work, we have focused only on 5-methylcytosine (5mC) as it is the most studied epigenetic biomarker in cancer. 5-hydrossymethylcytosine (5hmC) is another common epigenetic marker with promising application in cfDNA analyses [11,12]. Advances in this topic have been extensively reviewed elsewhere [13,14].

## 3. Pre-Analytics of cfDNA

cfDNA is a heterogeneous pool of fragmented DNA which includes genomic, mitochondrial host DNA, and DNA of bacterial origin [15]. It is a dynamic marker and can originate from different mechanisms of release such as cell death and active secretion processes [16]. cfDNA can be isolated from several body fluids such as urine, stool, airway, cerebrospinal fluid [17], and blood serum and plasma [18]. Individuals with cancer show an increased concentration of circulating nucleic acids with respect to healthy subjects due to the release of DNA, mRNA, and miRNA from tumor cells in the blood [19]. The tumor fraction of cfDNA is referred to as “circulating tumor DNA” (ctDNA). The use of ctDNA as a genetic marker is extremely challenging because of its fragmented nature, its limited yield, and the need to discriminate it from normal cfDNA [2]. Therefore, to move cfDNA analysis into clinical practice, there is a need to standardize pre-analytical conditions in order to preserve sample integrity and to obtain good quality data in downstream processes [20]. Pre-analytics of cfDNA samples consists of two main steps including biofluid collection and cfDNA isolation. As tumor DNA usually represents a small fraction of the total cfDNA, special consideration is required to avoid contamination by genomic DNA from other cells in order to optimize sensitivity and data homogeneity. The use of cell stabilizing tubes or quick sample processing when blood is collected in EDTA tubes is fundamental to obtain good quality samples [21,22]. For cfDNA isolation, there are several commercial kits capable of reaching good quality standard in terms of both DNA recovery and size distribution [23]. To avoid cfDNA degradation, samples should be stored at -80°C, although reduced cfDNA yield is reported for long-term storage [24]. Optimal pre-analytical guidelines for analyzing genetic and epigenetic information from cfDNA samples, including collection, quality control, centrifugation, storage, and DNA isolation have been recapitulated by Meddeb et al. [25].

## 4. DNA Treatment

DNA treatment is required as direct detection of DNAm is currently limited to only a few low-throughput techniques, such as nanopore and single molecule sequencing [26,27,28]. For this reason, the first key step for the analysis of DNAm in cfDNA samples is treatment of DNA that permits the use of DNA reading techniques such as PCR, microarray, and sequencing technologies. The main methods for DNA treatment are restriction enzyme (RE) digestion, bisulfite treatment, affinity enrichment methylated DNA, and methods combining enzymatic and chemical modification procedures.

RE digestion involves the use of methylation-sensitive enzymes, that cut unmethylated DNA only, leaving the methylated DNA intact, or methylation-insensitive enzymes [29,30]. RE digestion can be followed by PCR [31,32], microarray [33], or sequencing [30,34,35] based assays. DNA digestion by RE is a low-cost and simple method, but is error prone and is limited to providing information for the enzyme specific CpG sites only, thus limiting the applicability to the typically highly-fragmented cfDNA samples.

Sodium bisulfite treatment [36] represents the gold standard of DNAm detection, providing qualitative and quantitative estimation of DNAm at single pair resolution [37,38]. This method converts unmethylated cytosines to uracile residues that are represented as thymines on the amplified sense strand. In contrast, 5mC residues remain unaffected and are represented as cytosine residues on the amplified sense strand. After conversion, 5mC can be detected through PCR, microarrays, or sequencing based methods. This treatment is inexpensive, reproducible, and minimally time-consuming. One important limitation of bisulfite conversion is DNA degradation due to harsh chemical conditions such as low pH, high temperature, prolonged incubation times, high bisulfite concentrations, and alkali treatment [37,39,40]. Other limitations include incomplete conversion efficiency and over-conversion of 5mC due to DNA quality and quantity, purification procedures, chemical conditions, and the presence of resistant conversion sequences inducing false positive and false negative signals [41]. Moreover, this method is not able to distinguish 5mC from 5hmC, because both are resistant to conversion to uracil [42].

Affinity enrichment methods are based on either 5mc-specific antibodies (MeDIP) or methyl-binding domain (MBD) proteins to enrich for methylated DNA. In MeDIP, DNA is sonicated (not required when working with cfDNA) in fragments ranging from 300 to 1000 bp, denaturated, and then immunoprecipitated with monoclonal antibody against 5mC [43]. The immunoprecipitated DNA can be then analyzed with PCR, array, and sequencing-based techniques with resolution of about 100 bp in length [44]. Recently, Shen et al. developed a protocol (cfMeDIP-seq) by modifying the original MeDIP-seq, which is able to analyze the methylome starting from 1–10 ng of DNA input amount [45,46]. As such, this protocol is therefore suitable for liquid biopsy analyses. The major advantage of this protocol is the small DNA input required and its applicability to any fragmented or low-input DNA source, such as cfDNA from serum, urine, saliva, stool, cerebral spinal fluid, or ascites or genomic DNA from formalin-fixed paraffin-embedded tissue. MethylCap is a procedure based on capture of methylated DNA with the MBD domain of MeCP [47]. DNA is captured by a recombinant GST-MBD fusion protein and paramagnetic beads under low salt conditions obtaining the stratification of analyzed DNA into eluates with different methylated CpG density, which are subsequently sequenced [48]. Both the MeDIP and MethylCap methods detect only 5mC. MeDIP requires single-stranded DNA to select methylated DNA, while MethylCap binds double-stranded DNA, avoiding denaturation treatment. Regarding the binding affinity to methylated CpG sequences, MeDIP is able to enrich methylated regions with low CpG density more efficiently, while MethylCap tends to capture regions with higher CpG density [49].

Recently, two methods combining the use of enzymatic and chemical modifications have been developed: Enzymatic Methyl-seq (EM-Seq) and ten-eleven translocation (TET)-assisted pyridine borane sequencing (TAPS). EM-Seq (New England BioLabs, USA) uses the TET2 enzyme and an oxidation enhancer to protect modified cytosine from downstream deamination. TET2 oxidizes 5mC and 5hmC into 5-carboxycytosine (5caC), protecting these modifications from deamination. 5hmC can also be protected from deamination by glycosylation to form 5ghmC using the oxidation enhancer. The second step uses apolipoprotein B mRNA editing enzyme catalytic polipeptide-like (APOBEC) that deaminates cytosine but does not affect 5caC and 5ghmC. Converted sequences are the same as after bisulfite treatment and can be analyzed using similar downstream techniques. TAPS [50] combines TET oxidation of 5mC and 5hmC to 5-carboxylcytosine (5caC) with pyridine borane reduction of 5caC to dihydrouracyl (DHU). Subsequently, PCR converts DHU to tymine, inducing a C-to-T transition of 5mC and 5hmC.

## 5. Experimental Assays for DNAm Evaluation

Assays for DNAm evaluation can be generically divided into genome-wide and targeted methods (Figure 2). Genome-wide DNAm methods include whole genome bisulfite sequencing (WGBS), reduced representation bisulfite sequencing (RRBS), affinity enrichment sequencing, and microarray assays. Targeted methods rely on the detection of DNAm signal in pre-selected regions of interest. PCR-based techniques (PCR or ddPCR) are used to investigate single sites, while target sequencing enables the simultaneous analysis of multiple genomic regions, ranging from Kb to Mb in size. Although all the assays for DNAm evaluation were originally developed for the analysis of high-quality samples, in the section below, we report on methods that have been already applied to cfDNA samples or to low-quality samples, therefore representing assays of potential use in cfDNA analyses. A description of such methods along with a focus on experimental protocols amenable to cfDNA samples is reported in the sections below. A summary description of the analytical workflows of these assays is reported in Appendix A.

### 5.1. Whole-Genome Bisulfite Sequencing (WGBS)

The gold standard for genome-wide single-base resolution measurements of DNAm levels is WGBS. The first human genome-wide, single-base-resolution DNAm map by WGBS was produced in 2009 [51], with this technique having been extensively used both in basic and in translational research [52,53]. The major advantage of WGBS is its ability to evaluate the methylation state of nearly every CpG site, including low-CpG-density regions, such as intergenic regions, partially methylated domains, and distal regulatory elements [54]. Its disadvantages consist of the high degree of bisulfite induced DNA degradation, the uneven representation of thymine and cytosine bases, the high depth of sequencing required, and consequently, the very high costs of sequencing. Standard WGBS protocols require the adaptor to tag both ends of DNA fragments before bisulfite conversion to guarantee the amplification of pre-ligated DNA. This approach requires high input DNA. Technical improvements, such as single tube library preparation, the employment of bead-purification and the setup of different methods for library construction allow WGBS analysis with low or very low DNA input [55], such as in the context of cfDNA analyses [52,56,57,58]. Another WGBS approach on low input DNA is the post bisulfite adaptor tagging technique (PBAT). PBAT is a highly efficient PCR-free method that circumvents bisulfite induced DNA library degradation attaching adaptors after bisulfite treatment. This protocol can generate a PCR-free library starting from 125 pg of DNA [59,60]. PBAT protocol has been recently modified to evaluate methylation status at single cell level by single cell bisulfite sequencing (scBS-seq) [61,62]. Given these characteristics, both PBAT and scBS-seq are potentially applicable in cfDNA analyses.

TAPS and EM-Seq are sequencing methods based on enzymatic conversion of DNA (see previous section). EM-Seq has been used to develop cfNOMe (cell-free DNA-based nucleosome occupancy and methylation profiling), a single assay to evaluate the methylation status and nucleosome position on cfDNA from plasma and urine samples [63]. TAPS has also been successfully applied for cfDNA analysis [50]. It allows accurate long-range methylation sequencing providing higher quality data with reduced cost. Moreover, it is compatible with a variety of downstream analysis such as PCR, microarray, and sequencing based assays, including third-generation sequencing technologies, such as Nanopore and PacBio Single-Molecule Real-Time [64].

### 5.2. Representative Genome-Wide Methods

DNA methylation occurs predominantly in CpG dinucleotides which represent about 1% of total nucleotide in a vertebrate’s genome. RRBS selects for genomic regions rich in CpG dinucleotides, reducing sequencing throughput requirements. It therefore has limited applicability in detecting methylation changes in CpG-sparse regions including intergenic regions overlapping with functional enhancers [65]. RRBS combines the use of methylation restriction enzymes (MRE) to select for CpG-rich sequences with bisulfite sequencing. The use of two enzymes like *Msp*I and *Ape*KI enables the coverage of around 20% of CpG rich regions and increases the accuracy of sequencing compared to the single-enzyme RRBS method [66]. Several methods have been established in applying RRBS to low input DNA, such as cfDNA. The use of laser capture microdissection (LCM-RRBS) has reduced the need of DNA input to around 1 ng [67]. scRRBS is a modification of RRBS that allows the analysis of the methylation profile at a single cell level [68], which has been also used for the analysis of cfDNA samples from patients with lung and colorectal cancer [69]. To avoid DNA loss, scRRBS performs the following in a single tube: adapter ligation, DNA fragmentation by restriction enzyme, and bisulfite conversion of the ligated DNA. Purified DNA is then amplified and sequenced. Methylated CpG tandems amplification and sequencing (MCTA-seq) is another highly sensitive method capable to detect hypermethylated CpG islands in cfDNA samples [70]. This technique uses a single-tube three-step amplification of the DNA fragments adjacent to the methylated CGCGCGG sequences from bisulfite-treated DNA. The advantage of the MCTA-seq technique is the extremely low input of DNA required (about 7.5 pg). However, similar to RRBS, MCTA-seq can only detect CpG tandem regions missing non-CpG methylation.

### 5.3. Affinity Enrichment Sequencing

Affinity enrichment using monoclonal antibody specific for 5mC is a traditional method for studying epigenomics. Taiwo et al. demonstrated the feasibility of combining MeDIP with massively parallel sequencing starting from at least 50 ng of DNA input [45]. This protocol was then optimized by Shen et al. to assess genome-wide methylation profiles of plasma DNA from patients with different tumor types (cfMeDIP–seq) [71]. cfMeDIP–seq is based on the use of exogenous lambda DNA as a filler to increase initial DNA input. The filler works as a carrier for the immunoprecipitation reaction, increasing the specificity and enabling the maintenance of a constant antibody/DNA ratio without interfering with the analysis of sequencing data. cfMeDIP-seq can be used with low input (1–10 ng) and fragmented DNA. Similar to other immunoprecipitation-sequencing techniques, this approach is not able to quantify methylation at single-base resolution. However, it can recapitulate the same methylated regions obtained through WGBS or RRBS, but at a lower cost [46].

Methylated DNA fragments can also be enriched by using the MBD domain of methyl-CpG binding proteins coupled with magnetic beads and sequenced by methyl-CpG binding domain protein capture sequencing (MBD-seq) [48]. Recent optimization of this protocol enabled the use of less than 5 ng of DNA with 90% of sensitivity and a specificity similar to WGBS [72], thus making this technique suitable for cfDNA studies.

### 5.4. Microarrays

Microarray-based methods have been extensively used due to their low cost, ease of use, and high genome coverage. Hybridization can follow the digestion of DNA with methylation sensitive enzymes (MSREs), MeDIP, MBD, or bisulfite conversion [73]. Two examples of microarrays are CHARM (comprehensive high-throughput arrays for relative methylation) [74] and Illumina Infinium array (Illumina, San Diego, CA, USA). The Illumina Infinium Human Methylation 450 Bead Chip array (HM450) has been one of the most utilized array-based methods, able to quantify DNAm status of more than 485,000 CpG dinucleotides in 99% of known genes and 96% of CGIs at single base resolution [75]. It has been used in The Cancer Genome Atlas Consortium (TCGA) as the reference DNAm platform [76]. It has also been applied to plasma cfDNA to assess tissue-specific methylation patterns [21] or to identify biomarkers of abiteratone acetate resistance in patients with castrate- resistant prostate cancer [77]. The Illumina Infinium Methylation-Epic Bead Chip microarray is an improved version of HM450, covering more than 850 K CpG methylation sites. This array contains >90% of the HM450 sites plus 333,265 CpGs located in enhancer regions identified by the ENCODE and FANTOM 5 projects [78]. A modified MBD enrichment coupled with MeKL-chip (methylated DNA, kinase pre-treated ligation-mediated PCR amplification, and hybridization to the CHARM array) enabled the reduction of sample requirement to as little as 20 ng of fragmented DNA [79].

### 5.5. Targeted Sequencing

Targeted sequencing is a scalable high-throughput method which enables the evaluation of DNAm changes at single CpG resolution. Genomic target selection can be carried out by PCR amplification or by probe hybridization capture.

The first approach relied on genomic DNA bisulfite conversion, followed by multiplex amplification using specific and validated primers. Resulting libraries are then sequenced at deep coverage, allowing for sensitive detection of DNAm in low-input samples such as cfDNA. This approach, usually referred to as bisulfite amplicon sequencing, requires careful optimization to avoid biases in amplification yields of the different targets. For this reason, a limited number of sites are usually analyzed [80,81].

Targeted sequencing by probe hybridization enables the investigation of tens-to-millions of CpG sites in a single assay. The capture of the selected genomic regions relies on probe enrichment of libraries obtained from converted DNA. Several probe capture kits, such as Twist custom panels (Twist Bioscience, San Francisco, CA, USA) [82], Roche SeqCap Epi CpGiant Enrichment (Roche, Basel, Switzerland) [58], Illumina TruSight Rapid Capture Kit (Illumina, San Diego, CA, USA) [83], and Integrated DNA Technologies (IDT) custom panels (Integrated DNA Technologies, Coralville, IA, USA) [84,85] have been previously utilized for the analysis of DNAm from cfDNA samples. Focusing on specific regions of interest, target sequencing is cost-effective and shows improved sensitivity due to the degree of high sequencing depth that can be achieved.

### 5.6. Methylation-Specific PCR and Droplet Digital PCR

Methylation-specific PCR (MSPCR) is a sensitive method able to distinguish methylation status of target loci by specific primers designed on bisulfite converted DNA sequences or in MRE cleaved DNA. This approach has been extensively used to evaluate the methylation status of specific sites in cfDNA of patients with cancer [86], achieving a sensitivity of 0.1% of methylated alleles of a given CpG locus [76]. The design of specific and optimized primers is the most critical and challenging step to obtain adequate results using PCR-based methods. Several free software applications are available for primer design, including MSP-HTPrimer [87], PerlPrimer [88], and MethPrimer [89]. Due to the low tumor DNA concentration in cfDNA [21], the number of amplification cycles should be also carefully evaluated [90]. Nested PCR can be used to increase the assay specificity in DNAm analyses, and in particular, for low quality/low quantity specimens, such as cfDNA samples.

Quantitative real-time PCR assays have been developed to evaluate the DNAm status of sites of interest. MethyLight is a high-throughput quantitative real time PCR that utilizes a Taq-Man-based fluorescent probe to measure DNAm with a detection limit of 0.01% [91]. A multiplex MethyLight assay was applied to the serum cfDNA for the early detection of epithelial ovarian cancer [92]. A multiplex quantitative PCR, ligase reaction detection reaction quantitative PCR (LDRqPCR), has been set up to improve MSP on cfDNA and was shown to be able to evaluate seven CpG markers relevant in colon cancer [93,94]. Another PCR approach is methylation sensitive high-resolution melting (HRM) methods. HRM methods use differences between melting temperature of methylated and unmethylated PCR product to assess DNAm [95,96].

Droplet digital PCR (ddPCR) is an advance in PCR technology based on the generation of water-oil emulsion droplets in which converted DNA is entrapped and amplified. It is a quick method that enables a highly sensitive end-point PCR, providing absolute quantification of target sites in cfDNA. Epigenetic markers identified by ddPCR have been able to successfully discriminate patients with breast cancer from healthy volunteers [97], and can monitor treatment response in metastatic colorectal cancer [98]. ddPCR has also been applied to detect microRNA-34b/c methylation status in cfDNA in malignant pleural mesothelioma [99].

The advantages of PCR-based methods are their low-cost, ease-of-use, and accuracy. However, only a limited number of CpG sites can be evaluated in a single assay, and standardization efforts may be higher compared to other methods.

## 6. Computational Analysis of cfDNA Data

The analysis of the DNAm data is performed using assay-specific computational pipelines. In the case of PCR or ddPCR, analysis is usually carried out using vendor-provided software which produces quality metrics and absolute and/or relative estimations of DNAm. For microarrays and sequencing based assays, data analysis can be divided into two main steps, including data processing and downstream statistical analysis that is tailored to the aims of a given study. Computational methods for the analysis of microarrays and sequencing-based DNAm data are extensively reviewed elsewhere [100,101,102].

### 6.1. Data Processing

#### 6.1.1. Microarray

The processing of DNAm microarray data includes image processing, data normalization, and methylation calling and quantification. Image processing is performed using vendor-provided software (Illumina BeadScan, Illumina, San Francisco, CA, USA) that estimates, for each probe, intensity and absolute DNAm levels. To mitigate biases relative to the typical uneven distribution of probe intensity levels, commercial software (GenomeStudio, Illumina, San Francisco, CA, USA) or open-source packages that implement algorithms for signal normalization and background subtraction exploiting control probes can be used [103]. Specific methods, such as ComBat [104] and isva [105], allow users to diminish possible batch effects [102]. After normalization, the percentage of methylation for each site, referred to as “β-values”, is calculated.

#### 6.1.2. Bisulfite Sequencing Data

In the case of bisulfite sequencing data, the typical workflow for the analysis of high-throughput sequencing data is used. First, incorrectly converted reads are removed, and reads that have residual adapter sequences are trimmed using specific tools [106,107]. The next key step involves the mapping of the reads against a human reference genome using specific aligners, such as Bismark [108] and BSMAP [109], taking into account the characteristics of DNAm data. In particular, since sequencing reads derive from bisulfite treated genomes, an equivalently converted version of the reference genome (wild-card or three letter genome, see Bock et al. [102] for more details) is used to perform the mapping step. The outputs of these tools are alignment files, called BAM (binary sequence alignment map) files, that include the position of each read with respect to the reference genome. Methylation calls are then generated using tools, such as Bismark and Bis-SNP [110] that output specific genomic interval files (similar to browser extensible data (BED) files). These files report, for each queried genomic position, the number of reads supporting unmethylated (C) and methylated (T) CpG sites. The percentage of methylation (β-value) is also reported. To obtain reliable estimation of β-values, only sites covered more than a certain threshold are considered for downstream statistical analyses.

#### 6.1.3. Affinity Enrichment Sequencing Data

Differently from bisulfite sequencing providing DNAm information at single site level, affinity enrichment assays, such as MRE-seq and MeDIP-seq, measure the enrichment or depletion of sequencing reads that map to specific genomic regions. First, mapping of the reads with standard short read aligners, such as BWA [111] or Bowtie2 [112] is performed. After normalization for the uneven CpG distribution in the genome [44], the region-level relative enrichment or depletion of DNAm can be estimated through specific tools, such as MeDUSA [113] or MEDIPS [114].

### 6.2. Statistical Analysis of Cell-Free DNAm Data

The statistical analysis of DNAm sites and/or regions from microarrays or sequencing data can be performed using different tools [101] developed to identify differentially methylated sites and/or regions across two or multiple conditions [115,116], to estimate tumor purity [117,118] and stromal and immune cell admixture [119], or to identify differentially variable sites and/or regions [120]. However, most of these tools are designed for the analysis of tissue samples, and there are no reference methods that take into account the peculiarities of cfDNA data. In fact, as previously mentioned, tumor DNA is mixed (usually at low proportion) with the cell-free normal DNA, thus requiring ad-hoc analytical strategies depending on the clinical questions. For these reasons, different analytical strategies are provided below as illustrative examples.

Gordevičius et al. [77] used HM450 to generate the cfDNA methylation (cfDNAm) profiles of 108 plasma samples collected from 33 patients with advanced prostate cancer. After processing data with Minfi [103], a standard pipeline for the analysis of microarrays, a mixed effects linear interaction model was applied to identify significantly differentially methylated sites in patients with drug resistance versus sensitivity.

Beltran et al. [52] generated WGBS of 11 patients with prostate cancer and studied whether cfDNA-methylation was able to differentiate between androgen-receptor (AR)-indifferent and AR-dependent advanced prostate tumors. High concordance between cfDNA-methylation and DNAm signal detected in available matched tissue samples was observed. Unsupervised hierarchical clustering on known differential DNAm sites (AR-indifferent vs. AR-dependent) well recapitulated the DNAm signal observed in tissue samples. Of note, Purity Assessment from clonal MEthylation Sites (PAMES) [117]—a tool we developed to estimate tumor admixture in tissue sample data—provided tumor fraction estimates concordant with CLONality Estimate in Tumors (CLONET) [121], a whole exome sequencing (WES)-based method applied to available matched WES data.

To demonstrate the feasibility of non-invasively detect cancer in patients with different cancers, Liu et al. [83] developed a pan-cancer panel including about 10k CpG sites and applied it to the cfDNA of 78 patients. Using 20 normal samples as reference for mean and standard deviation, a Z-score statistics was developed to inform about the presence of cancer. In Panseer [122], another target sequencing approach including about 12k CpG sites across 595 genomic regions, the authors first computed the average methylation fraction (AMF) across each targeted genomic region and then build a logistic regression model to classify samples from healthy patients or patients with cancer.

As reported in previous sections, Shen et al. [71] introduced cfMeDIP-seq to generate region-wise DNAm profiling of cfDNA samples. In this study, in order to detect differentially methylated regions (DMRs) and quantify circulating tumor content, a Z-score-based strategy was applied to hundreds of samples from patients with different cancers and healthy donors. Using the same assay to identify subtypes in intracranial tumors, Nassiri et al. [123] used a random forest model trained on the top 300 DMRs for gliomas, common extra-axial tumors, and intra-axial tumors versus each other class, demonstrating the high accuracy of their approach. Similarly, Nuzzo et al. trained a GLMnet classifier [124] on the top 300 DMRs to distinguish between individuals with renal cell carcinoma, without cancer and amongst different genitourinary tumors [125].

## 7. Applications of Cell-Free DNA-Methylation Assays in Oncology

### 7.1. Tumor Detection and Monitoring

The measurement of DNAm-based biomarkers in cfDNA samples holds great potential for noninvasive detection of cancer, both in early and advanced disease [20]. In the early setting, the evaluation of minimal residual disease after surgery using DNAm in liquid biopsy may improve the discrimination of patients who would benefit the most from adjuvant treatment [126]. In the advanced setting, disease status can be monitored with sequential measurements of cfDNA to quantify circulating tumor content and to evaluate the response to therapy, thus enabling early identification of patients with disease resistant to treatment. For these reasons, many studies have been conducted to identify DNAm-based biomarkers for tumor detection and monitoring.

A common approach relies on the identification of DNAm markers exploiting publicly available data on cancer and normal tissue to nominate a reduced set of promising candidates. Targeted techniques, such as target bisulfite sequencing and/or PCR-based assays, are then applied. Using this strategy, Luo et al. developed a diagnostic model (cd-score) using nine cfDNA methylation markers to discriminate patients with colorectal cancer (CRC) from normal individuals. The cd-score showed a higher sensitivity than serum carcinoembryonic antigen (CEA), a widely utilized tumor marker. Futhermore, it correlated with the staging of CRC and with the presence of residual disease after treatment and response to treatment. In a high-risk population of 1493 participants in a prospective cohort study, they also identified a single DNAm marker (cg10673833) to accurately detect patients with cancer and precancerous lesions [84]. In a similar study, Xu et al. developed a 10 DNAm marker panel to discriminate patients with hepatocellular carcinoma (HCC) from individuals with HBV/HCV infection, fatty liver disease and healthy controls. The sensitivity of the score derived from this assay was both comparable to liver ultrasound and superior to alpha-fetoprotein, the only available test for HCC detection and surveillance, showing high correlation with tumor burden and treatment response [85]. In the prospective study TBCRC 005, Visvanathan et al. applied the quantitative multiplex methylation-specific PCR assay “cMethDNA” [127] to the serum samples from 141 patients with metastatic breast cancer at baseline, at 4 weeks, and at first restaging. The cumulative methylation index (CMI) based on the DNAm level of the six genes included in this assay was able to stratify patients based on both progression-free survival (PFS) (2.1 months for high CMI vs. 5.8 months) and overall survival (OS) (12.3 months vs. 21.7 months). Additionally, an increase in the CMI from baseline to week 4 was associated with worse PFS and likelihood of progressive disease at first restaging [128]. Recently, Chen et al. reported PanSeer, a targeted approach to detect five common cancers (stomach, esophagus, colorectal, lung, and liver) which was able to detect cancer in 95% of asymptomatic individuals who were diagnosed 4 years later [122]. The results of the application of another multi-cancer targeted methylation assay to a cohort of 6689 participants, including 2482 patients with cancer, were presented by Liu et al., demonstrating the feasibility of detecting >50 cancer types across all stages of disease, albeit showing less sensitivity for lower (I-III) stage tumors [82].

The other strategy exploiting DNAm-based biomarkers for tumor detection and monitoring is based on the analysis of cfDNA samples via genome-wide techniques, such as microarrays and sequencing based assays, followed by the application of statistical learning methods for the identification of candidate markers. As already mentioned, many groups have exploited cfMeDIP-seq to assess the genome-wide methylation profiling of cfDNA from patients with different cancer types [71], leading also to the identification of a set of biomarkers with potential clinical application [129]. cfMeDIP-seq has been used to detect metastatic renal cell carcinoma, resulting in more sensitivity than DNA-based assays (gene panel) [130]. Recently, Nassiri et al. [123] and Nuzzo et al. [125] showed the feasibility of cfMeDIP-seq in discriminating between distinct subtypes in intracranial tumors and renal cell carcinomas, respectively (see previous sections for more details).

Using target bisulfite sequencing and low-pass whole-genome bisulfite sequencing, Wu et al. conducted a study to characterize the metastatic castrate-resistant prostate cancer (CRPC) methylome in cfDNA from men with CRPC treated with either abiraterone or enzulatamide in the pre- or post-chemotherapy setting. They observed that the main contributor to methylation variance was the hyper-methylation of polycomb repressor complex 2 component (PRC2) which was strongly correlated with tumor fraction. Subsequently, they defined a signature called AR-MethSig, which was able to identify subgroups of cancer with a more aggressive clinical course [58]. In another study, Wen and colleagues used MCTA-seq to detect thousands of hypermethylated CpG islands in cfDNA from plasma samples of patients with HCC. After the identification of candidate DNAm markers through a study of tissue and plasma samples, the authors selected four of them specific for cancer detection [70].

### 7.2. Drug Resistance

Despite the great improvements made in cancers therapy in the last decades, treatments resistance remains one of the major and ubiquitous problems in cancer management. Increased evidence showed that epigenetic modification plays a key role in determining tumor drug resistance [131,132,133]. However, to date, few studies have evaluated epigenetic markers of drug resistance in cfDNA samples.

Some studies have focused on the analysis of single DNAm biomarkers. Persistence or new appearance of serum *RASSF1A* was proposed as a circulating epigenetic marker of resistance to adjuvant tamoxifene therapy in patients with early breast cancer [134].

Using the genome-wide approach, Gordevičius and colleagues identified a list of 21 potential predictive biomarkers of abiraterone acetate resistance analyzing the cfDNA methylation profile of 33 patients with castrate-resistant prostate cancer [77]. Beltran et al. [52] analyzed cfDNA of patients with hormone-naive metastatic prostate adenocarcinoma, CRPC adenocarcinoma (CRPC-Adeno), and CRPC neuroendocrine prostate cancer (CRPC-NE) and compared their genomic and epigenomic profile with patient-matched tumor biopsies. A signature combining genomic (mutations involving *TP53*, *RB1*, *CYLD*, *AR*) and epigenomic alterations (top 20 hypo- and 20 hyper-methylated sites) was developed and tested in WGBS cfDNA samples to distinguish patients with CRPC-NE from those with CRPC-Adeno.

### 7.3. Cell-of-Origin Identification

The identification of the cell-of-origin of tumors from cfDNA is of great importance for improving the diagnosis, treatment, and prognosis in cancer. This is particularly relevant for patients with cancer of unknown primary (CUP) who could benefit from site-specific therapies. In this context, DNAm is an ideal marker as it harbors marked cell-type-specific information [76].

A common strategy for the identification of cancer-initiating cell-type is to use the large amount of data generated by The Cancer Genome Atlas (TCGA) consortium to first identify cancer-specific DNAm-based markers. Moss et al. [21] applied this concept to the cfDNA-methylation profiling of patients with metastatic colon, lung, and breast cancer using a computational method that deconvolutes patient cfDNA to infer cell-type composition. Similarly, CancerLocator is a probabilistic method to identify cell-of-origin based on cfDNA methylation sequencing, showing superior performance compared to existing state-of-the-art multi class classification methods [135].

A different approach was utilized by Guo et al. [69] which extended the concept of linkage disequilibrium to the analysis of CpG patterns and defining a metric, called methylation haplotype load, to infer tissue-specific methylation signal. Applying this method to the RRBS of cfDNA from 59 patients with lung or colorectal cancer, they demonstrated quantitative estimation of tumor load and accurate identification of cell-of-origin. The identification of cell-of-origin through methylation haplotypes was also exploited by Li et al. [136] which introduced a new probabilistic method, named CancerDetector suitable for data generated from low-to-medium coverage (1× to 10×) assays (i.e., WGBS and RRBS), having the potential to cost-effectively detect cancer early. The possibility of detecting cell-of-origin was also investigated by Liu et al. with a tailored targeted methylation assay capable of localizing the tissue of origin with >90% accuracy [82].

## 8. Clinically Approved DNA-Methylation Assays

To date, several DNAm-based biomarkers have been proposed, but relatively few have been included in clinical guidelines and approved for liquid biopsy application [137] (Table 1).

In the context of colorectal cancer, the Cologuard is a Food and Drug Administration (FDA, USA) approved test (Exact Sciences Corporation, Madison, WI, USA) for general CRC screening of average-risk adults older than 50 years old. The Cologuard IVD is a stool-based DNA test based on the analysis of the DNAm level of *NDRG4* and *BMP3*, the mutational status of *KRAS* gene and on hemoglobin immunoassay. In a study involving 9989 subjects, this assay showed high sensitivity and specificity (92% and 87%, respectively) for CRC detection and higher sensitivity than commercially available fecal immunochemical tests (FIT) for human hemoglobin in the detection of both colorectal cancer and advanced precancerous lesions, albeit with lower specificity (87% vs. 95%) [138]. In the context of these results, the Cologuard test may be considered as a potential alternative to colonoscopy [144]. The Epi proColon 2.0 test (Epigenomics AG, Berlin, Germany) was approved by FDA in 2016 as the first blood test for early CRC detection based on analysis of the methylation status of the *SEPT9* gene on plasma samples. This test shows high sensitivity (81%) and specificity (97%) [139]. The Epi proColon 2.0 test is not recommended for routine screening of CRC, but it is a sensitive screening option for patients >50 years old with average risk for CRC who decline CRC screening by colonoscopy or FIT [139].

The Epi proLung (Epigenomics AG, Berlin, Germany) is a CE-IVD diagnostic tool for the detection of lung cancer, analyzing the methylation status of *PTGER4* and *SHOX2* on cfDNA. This test can be considered as a complementary tool to current screening methods to improve the selection of individuals eligible for low dose computed tomography screening [140,145]. Epigene (Epigene, iStat Biomedical Co., New Taipei City, Taiwan) has outlined the Cervi-M and Oral-M DNA assays for the diagnosis of cervical and oral cancer. These assays are CE-IVD tests based on the analysis of the DNAm level of *ZNF582* and *PAX1* from oral swab or cervical brush samples [141,142]. Assure MDx (MDxHealth, Irvine, CA, USA) is another CE-IVD assay designed to evaluate the risk of bladder cancer in patients with hematuria rather than relying on cystoscopy [146]. This assay combines the analysis of the DNA-methylation of *TWIST1*, *ONECUT2*, and *OTX1* and mutational status of *FGFR3*, *TERT*, and *HRAS* from urine samples [143].

## 9. Conclusions

Liquid biopsy holds great promise in the management of patients with cancer. Several studies have demonstrated the utility of cfDNA analysis for tumor detection and monitoring, personalized treatment of patients with advanced cancer, and identification of the tissue of origin of a tumor. In this review, we have focused on recent advances in experimental and computational methodologies for the analysis of DNAm from cfDNA samples, including sample collection and pre-processing, DNA treatment, assays for DNAm evaluation, and data analysis. A summary of the characteristics of these experimental strategies, along with practical consideration for their usage, including minimum sample requirement and cost, is reported in Table 2.

In recent years, the analysis of DNAm in cfDNA of patients with cancer has gained much attention. Recent technological developments are assisting to establish the use of DNAm-based liquid biopsy tests in oncological practice. DNAm shows important advantages compared to genomics-based analysis as it is more prevalent, being markedly recurrent within a given tumor type, and pervasive, affecting a substantial fraction of cancer cell genome [117]. DNAm is extremely cell-type-specific [76], carrying relevant information about the cell of origin of a tumor. In addition, DNAm generally reflects dysregulated gene expression patterns and could be therefore considered as a surrogate of gene expression signatures. In the last two decades, several clinically valuable gene expression signatures have been developed to inform about pathways deregulated in cancer [149], tumor subtypes [150,151], actionable genomic alterations and drug resistance [152,153]. Due to its characteristics, we envision that DNAm will enable the evaluation of gene expression signatures in cfDNA samples, strengthening the field of translational epigenetics in oncology.

To conclude, given many recent successful works, we foresee that future advancements in biochemical and computational methodologies for the analysis of DNAm from cfDNA will overcome current technical challenges, providing improved detection power and reproducibility and thus establishing the applicability of DNAm-based cfDNA analyses in oncology.

## Figures and Tables

**Figure 1 biomolecules-10-01677-f001:**
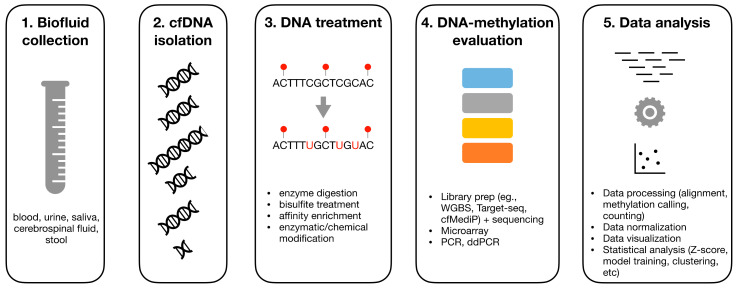
Schematic representation of a typical DNA-methylation analysis workflow from cell-free DNA samples including pre-analytics (biofluid collection and cfDNA isolation), DNA treatment, DNA-methylation evaluation, and computational data analysis.

**Figure 2 biomolecules-10-01677-f002:**
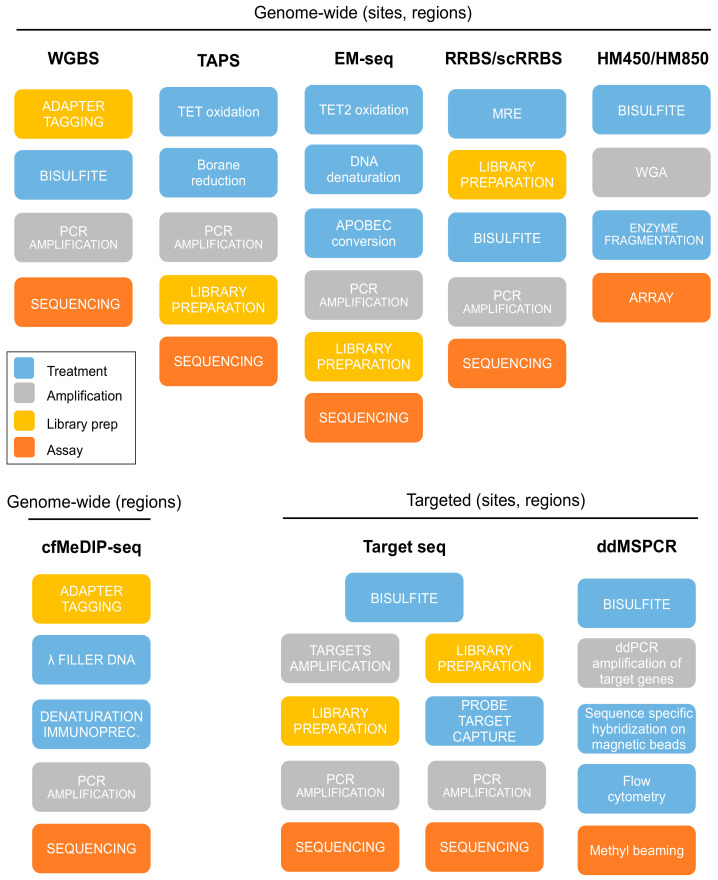
Summary of the experimental assays for DNA-methylation evaluation. Genome-wide techniques include methods able to perform site and region wise analysis, such as whole genome bisulfite sequencing (WGBS), TET-assisted pyridine borane sequencing (TAPS), enzymatic methyl-sequencing (EM-seq), reduced representation of bisulfite sequencing (RRBS, single-cell RRBS (scRRBS)), and infinium methylation arrays (HM450, HM850), and methods analyzing DNAm regions only, such as circulating free methylated DNA immunoprecipitation sequencing (cfMeDIP–seq). Targeted methods include target bisulfite sequencing (target bisulfite seq) and PCR-based assays, including methylation specific PCR (MSPCR) and droplet digital methylation specific PCR (ddMSPCR). MRE: methylation restriction enzyme, WGA: whole genome amplification.

**Table 1 biomolecules-10-01677-t001:** Characteristics of clinically approved DNA-methylation assays for the detection of different type of cancers.

Name	Manufacturer	Biomarker(s)	Biosample	Application	Sensitivity (%)	Specificity (%)	Reference
Cologuard	Exact Sciences Corp.	*NDRG4, BMP3*	Stool	CRC early detection	92	87	[138]
Epi proColon 2.0	Epigenomics GA	*SEPT9*	Plasma	CRC early detection	81	97	[139]
Epi proLung	Epigenomics GA	*PTGER4, SHOX2*	Plasma	Lung cancer detection	90	73	[140]
Cervi-M	Epigene, iStat Biomedical Co	*ZNF582, PAX1*	Cervical brush	Cervical cancer detection	77 *, 70 **	87 *, 82 **	[141]
Oral-M	Epigene, iStat Biomedical Co	*ZNF582, PAX1*	Oral swab	Oral cancer detection	85 *, 72 **	87 *, 86 **	[142]
Assure MDx	MDxHealth	*TWIST1, ONECUT2, OTX1 (+FGFR3, TERT, HRAS* mutations)	Urine	Bladder cancer detection	97	83	[143]

* Value refers to *ZNF582*, ** value refers to *PAX1.*

**Table 2 biomolecules-10-01677-t002:** Characteristics of the experimental assays suitable for the analysis of DNA methylation in cell-free DNA samples, including sample requirement, DNA treatment protocols, summary of advantages and disadvantages of the assays, cost and references to relevant studies.

	Assay	Sample Requirement [Minimum]		DNA Treatment				Advantages	Disadvantages	Cost	Reference
				**Bisulfite Conversion**	**Restriction Enzyme**	**DNA Precipitation**	**Enzymatic Plus Chemical Modification**				
Genome wide	Microarray	HM450	500 ng	**■**				pre-designed panel, easy to use, time efficient	DNA degradation; low coverage of intergenic regions	••	[21,77,78,147]
		HM850	250 ng	**■**				As above; includes enhancer regions; suitable for FFPE DNA;	As above	••	
		MeKL-chip	10–20 ng			**■**		low DNA input	Cross hybridization, PCR amplification, MBD binding ability	••	[79]
	Whole genome sequencing	WGBS	10 ng	**■**				full methylome	DNA degradation; requires high sequencing depth; low input DNA may induce PCR bias	•••••	[51,53,54,56]
		PBAT	125 pg–10 ng	**■**				full methylome; PCR free; suitable for single cell analysis	DNA degradation; requires high sequencing depth; low fraction of aligned reads	•••••	[59,60,61,62]
		TAPS	1 ng				**■**	no DNA degradation; low input DNA; suitable for third generation sequencing; detect 5mC and 5hmC	hyper-active TET1 preparation	••••	[63,64]
		EM-seq	100 pg				**■**	no bisulfite DNA degradation; very low DNA input; high mapping; uniform GC coverage; detect 5mC and 5hmC	low complexity sequencing library	•••	[50,63]
	Representative genome wide methods	RRBS	10–100 ng	**■**	**■**			high CpG coverage	DNA degradation, low coverage of intergenic regions	•••	[66,67]
		scRRBS	one cell	**■**	**■**			very low DNA input		•••	[68,69]
		MCTA-seq	7.5 pg	**■**				very low DNA input	DNA degradation, low coverage of intergenic regions	•••	[70]
		cfMeDIP–seq	1–10 ng			**■**		no bisulfite DNA degradation, no mutation introduced, genome wide CpG and no CpG, very low DNA input	Detect only regions, low GpG density bias, CNA bias, depending on antibody performance	•••	[45,71,123,125]
		MBD-seq	5 ng			**■**		no bisulfite DNA degradation, outperform meDIP-seq in regions with higher CpG density	hypermethylated regions bias, CNA bias, depending on antibody performance	•••	[48,72]
Targeted	Sequencing	Target bisulfite seq	20-30 ng	**■**				high coverage on target loci	primer or probe design	••/•••	[58,80,81,82,83,84,85,122]
	PCR	MSPCR	Pg	**■**	**■**			low DNA in input; relative quantification of target loci	primer or probe design	•	[86,91,92,94,95,148]
		ddMSPCR	pg	**■**	**■**			low DNA input; absolute quantification of target loci	primer or probe design; quantification depends on DNA input	•	[97,98,99]

HM450: Infinium Human Methylation 450 Bead Chip (Illumina), HM850: Infinium Methylation EPIC BeadChip Kit (Illumina), MeKL-chip: methylated DNA, Kinase pre-treated ligation-mediated PCR amplification and hybridization to the CHARM array, WGBS: whole genome bisulfite sequencing, PBAT: post bisulfite adaptor tagging technique, TAPS: TET-assisted pyridine borane sequencing, EM-seq: Enzymatic Methyl-Seq, RRBS: Reduced representation of bisulfite sequencing, scRRBS: single cell reduced representation of bisulfite sequencing, MCTA-seq: Methylated CpG tandems amplification and sequencing, cfMeDIP–seq: circulating free methylated DNA immunoprecipitation sequencing, MBD-seq: methyl-CpG binding domain protein capture sequencing, Target bisulfite seq: target bisulfite sequencing, MSPCR: methylation specific PCR, ddMSPCR: droplet digital methylation specific PCR. The number of filled circles schematically represents the cost of the assays, from cheaper (•) to more expensive (•••••) assays.

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
