# Peer review of "Cell-Free DNA-Methylation-Based Methods and Applications in Oncology"

_biomolecules, 2020, doi:10.3390/biom10121677_

Round 1
Reviewer 1 Report
In this review, Galardi et al. report on advances of analysis of DNAm from cfDNA samples. They describe DNAm analysis workflows, methods for data analysis, protocols, computational strategies as well as practical considerations on input requirements and costs. They also provide an overview on studies exploiting cell-free DNAm cancer biomarkers, drug resistance, the cell-of-origin of tumors and report on DNAm-based tests approved for clinical use.
This is a comprehensive, informative, very well structured, concise and well written review. The authors provide the reader an excellent overview of current advances of DNAm-based methods from cfDNA.
Author Response
We thank the Reviewer for his/her positive comments on our manuscript.
Reviewer 2 Report
This review manuscript summarized current methods for detecting DNA methylation in clinical and basic research, and provided a comparison of the advantages and limitations of each method. This review is focusing on cfDNA, which could be used as biomarkers on cancer diagnosis and monitoring, and on the evaluation of drug resistance. The table in this review provided very helpful information. In sum, this review is comprehensive and informative, from the view of sample categories, DNA treatment and data analysis.
Major concerns,
- As the authors mentioned that cell-free DNA (cfDNA) fragments could be detected in blood, urine, saliva, cerebrospinal fluid and stool, however, the authors did not describe and summarize the methods that used to isolate and purify cfDNA from liquid biopsy. Please add a section.
- This review manuscript is comprehensive, which includes almost all the current methods. However, there are too many similar terms described, which are hard to be followed. It will be helpful to provide a table to group the similar methods together by showing the differences and modifications, and emphasize and describe one specific method in detail in text.
- For the section of data processing, the description is too simply and hard to be followed. The authors should expand this section.
Author Response
This review manuscript summarized current methods for detecting DNA methylation in clinical and basic research, and provided a comparison of the advantages and limitations of each method. This review is focusing on cfDNA, which could be used as biomarkers on cancer diagnosis and monitoring, and on the evaluation of drug resistance. The table in this review provided very helpful information. In sum, this review is comprehensive and informative, from the view of sample categories, DNA treatment and data analysis.
- We thank the Reviewer for his/her positive comments on our study. We would be happy to provide the community with a comprehensive review on this emerging field and also to provide the readers with practical considerations on available methods and experimental strategies.
Major concerns,
- As the authors mentioned that cell-free DNA (cfDNA) fragments could be detected in blood, urine, saliva, cerebrospinal fluid and stool, however, the authors did not describe and summarize the methods that used to isolate and purify cfDNA from liquid biopsy. Please add a section.
- We thank the Reviewer for pointing this out. To address this Reviewer’s concern, in the current version of the manuscript we have extended the section “3. Pre-analytics of cfDNA” including information about isolation of cfDNA from liquid biopsy samples. However, the aim of our work is to cover all the aspects of the entire processing of cfDNA samples, from collection to data analysis. For more details about the isolation and purification of cfDNA from liquid biopsy samples, as well as the other pre-analytical procedures, we have referred to a review focused on the pre-analytics of cfDNA samples (Meddeb et al., https://doi.org/10.1373/clinchem.2018.298323).
This review manuscript is comprehensive, which includes almost all the current methods. However, there are too many similar terms described, which are hard to be followed. It will be helpful to provide a table to group the similar methods together by showing the differences and modifications, and emphasize and describe one specific method in detail in text.
- We agree with the Reviewer that the descriptions of the many protocols reported in our work may be hard to follow. To address this Reviewer’s concern, the current version of the manuscript now includes a new Supplementary Table (Supplementary Table 1) summarizing the analytical workflows of all the protocols mentioned in the main text. To show common and peculiar analytical steps, the protocols were grouped by DNA pretreatment methods and by principal analytical steps (DNA treatment, library preparation, amplification, analysis). We think that this additional table could facilitate the comprehension of the many assays reported in the text and of their peculiarities.
Regarding the suggestion “describe one specific method in detail in text”, in this work we would like to provide the community with a comprehensive description of available methods to facilitate potential readers in the choice of the more appropriate method/protocol for their own studies.
For the section of data processing, the description is too simply and hard to be followed. The authors should expand this section.
- The section “6.1 Data processing” has now been extended following this Reviewer’s suggestion. For more details about this topic we have referred to a comprehensive review on DNA-methylation data analysis (Bock et al., https://doi.org/10.1038/nrg3273).
Reviewer 3 Report
This review barely give any information of the major challenging in cfDNA studies: extracting enough cfDNA for subsequent studies. It gave a lot description of various methods of DNA methylation assays. The title “cfDNA-methylation-based methods” is very misleading. The authors are overly optimistic of detecting mDNA by conventional methods with cfDNA, which in practice many of the listed methods are not able to be used in cfDNA studies. The authors also failed to characterize challenging of the cfDNA from DNA obtained from tissues/cells. For example, the cfDNA are short and fragmented with a shot half-life. In the section of 3. pre-analytics of cfDNA, totally ignored this and proceed as DNA with enough amount to be used in many methods.
For Figure 2, suggest spelling out all the abbreviations in the figure, it is hard to follow with all the abbreviations.
part 8, clinically approved assays, can list a table for easy reading.
some abbreviation such as APOBEC should spell out when 1st use it.
Author Response
This review barely give any information of the major challenging in cfDNA studies: extracting enough cfDNA for subsequent studies. It gave a lot description of various methods of DNA methylation assays. The title “cfDNA-methylation-based methods” is very misleading. The authors are overly optimistic of detecting mDNA by conventional methods with cfDNA, which in practice many of the listed methods are not able to be used in cfDNA studies.
- The aim of this work is to cover all the aspects of the analytical workflow of cfDNA samples, from sample collection, to sample processing and data analysis. In this work we reported only on methods that have been already applied in the context of the analysis of cfDNA samples or to low-quality/quantity samples and therefore potentially applicable to cfDNA analyses. As likely we failed to specify this in the text, to address this Reviewer’s point we have added the following sentence in the section “5. Experimental assays for DNAm evaluation”.
Although all the assays for DNAm evaluation were originally developed for the analysis of high-quality samples, in the section below we report on methods that have been already applied to cfDNA samples or to low-quality samples, therefore representing assays of potential use in cfDNA analyses. A description of such methods along with a focus on experimental protocols amenable to cfDNA samples is reported in the sections below.
The authors also failed to characterize challenging of the cfDNA from DNA obtained from tissues/cells. For example, the cfDNA are short and fragmented with a shot half-life. In the section of 3. pre-analytics of cfDNA, totally ignored this and proceed as DNA with enough amount to be used in many methods.
- In the current version of the manuscript we have extended the section “3. Pre-analytics of cfDNA” describing how samples are processed after their collection. For more details about this part we have referred to a review focused on the pre-analytics of cfDNA samples (Meddeb et al., https://doi.org/10.1373/clinchem.2018.298323).
For Figure 2, suggest spelling out all the abbreviations in the figure, it is hard to follow with all the abbreviations.
- The caption of Figure 2 now reports all the abbreviations reported in the corresponding figure.
part 8, clinically approved assays, can list a table for easy reading.
- The current version of the manuscript now includes the new Table 1 summarizing the assays reported in the section “8. Clinically approved DNA-methylation assays”. We would like to thank the Reviewer for this suggestion as this change has improved significantly the readability of this section.
some abbreviation such as APOBEC should spell out when 1st use it.
We carefully revised all the manuscript and spelt out all the abbreviations reported in the text.
Round 2
Reviewer 3 Report
This is an improved version, with more discussion of cfDNA specific characteristic and challenges. However, the authors should go through the text and add specific changes/challenges for cfDNA in the methods have been discussed. If such studies are not available, should mark out and indicate those are for conventional DNA methylation studies. As some of the methods, such as MSPCR need to do nested PCR even with DNA from the cells (due to the difficulty of obtaining good primers at CG rich sequence after conversion and the efficiency for bisulfite conversion).
Author Response
This is an improved version, with more discussion of cfDNA specific characteristic and challenges.
- We thank the Reviewer for her/his positive comment about our study.
However, the authors should go through the text and add specific changes/challenges for cfDNA in the methods have been discussed. If such studies are not available, should mark out and indicate those are for conventional DNA methylation studies. As some of the methods, such as MSPCR need to do nested PCR even with DNA from the cells (due to the difficulty of obtaining good primers at CG rich sequence after conversion and the efficiency for bisulfite conversion).
- Following this Reviewer’s suggestion, in the revised version of the manuscript we have improved the description of the methods that have been reported. In particular, in the section “5. Experimental assays for DNAm evaluation”, we have better specified which methods have been already applied for the analysis of cfDNA samples, referring to available studies and highlighting which features enable their use in the context of liquid biopsy analyses. Also, we better specified which of the mentioned methods have not yet been used for the analysis of cfDNA samples, but highlighting the features that make them potentially applicable for cfDNA studies.